# Study on the Spatial Interaction between Urban Economic and Ecological Environment—A Case Study of Wuhan City

**DOI:** 10.3390/ijerph191610022

**Published:** 2022-08-14

**Authors:** Liang Geng, Xinyue Zhao, Yu An, Lingtong Peng, Dan Ye

**Affiliations:** School of Science, Hubei University of Technology, Wuhan 430068, China

**Keywords:** ecological environment, multisource data fusion, decoupling elasticity coefficient method, spatial autocorrelation model

## Abstract

In order to study the interactive relationship between urban economic and ecological environment, taking Wuhan as an example, Landsat and MODIS remote sensing satellite data and social and economic data were fused with multisource data, and multidimensional indicators were selected to construct the comprehensive evaluation index system of urban economic and ecological environment. The weights were determined by combining subjective and objective methods. Then, the decoupling elasticity coefficient method and spatial autocorrelation model were used to evaluate the dynamic relationship and spatial relationship between economic development and ecological environment in Wuhan from 2014 to 2020. The results showed that there was an interaction between the urban economic and the ecological environment in Wuhan. The ecological level index had a spatial effect, the adjustment of industrial structure had a positive effect on the improvement of the ecological level, and the improvement of the ecological level was also helpful to promote economic development. The typical districts of Huangpi District, Xinzhou District, Jiangxia District, Hannan District, Caidian District, and Hongshan District had superior location and ecological advantages, as well as high development potential. Lastly, on the basis of the empirical analysis results, policy suggestions are made from four aspects: regional differentiated construction, green development, energy consumption, and wetland construction.

## 1. Introduction

Coordinated development of urban economic and ecological environment is an important way to achieve regional sustainable development [1,2]. Strengthening ecological construction plays an important role in enhancing ecosystem function and increasing the carbon sink increment, which is conducive to further promoting high-quality development of urban economy [3,4]. However, the ecosystem is affected by urban expansion, construction, and other social and economic activities, and the regional ecological environment and function will be destroyed [5,6]. Some international examples have also illustrated this problem. The study of Octavio et al. [7] showed that, since 2000, with the intensification of urban project development in Chile, the wetland area has gradually decreased, leading to the ecological fragility of urban areas and a reduction in disaster reduction capacity. Basit et al. [8] emphasized that the deterioration of the green belt infrastructure system in Peshawar City, Pakistan may have been caused by inappropriate development plans in urban fringe areas and insufficient maintenance of green structures. Cities in some countries still have slums and vulnerable neighborhoods in cities whose green infrastructure is harder to secure. Moreover, the studies of Powell and Lever [9] on Roma outskirts and Cretan et al. [10] and Méreiné-Berki et al. [11] on urban Roma confirmed that marginalized residents could face strong negative community impact at an urban level.

At present, the research on the relationship between urban economic and ecological environment is mainly based on the theory of urban ecological economic system [12,13,14]. The urban eco-economic system is an organic whole composed of the urban eco-system and the urban economic system, along with their interaction [15,16,17]. On the one hand, the pollutants produced in the process of economic development will harm the urban ecosystem; on the other hand, the urban ecosystem will slow down the process of urban economic development through resource shortage, environmental degradation, and other factors. Therefore, how to quantify the relationship between urban economic and ecological environment and promote the coordinated development of economy and the ecological environment is a hot issue worth paying attention to.

Starting from the basic principles of eco-city construction, Song [18] divided the main contents and ecological functions of the city, and studied the relationship between eco-city and sustainable development. Deng [19] analyzed the tradeoff between urbanization and ecological construction through land-use change. Chen [20] analyzed the spatial interaction between high-resolution remote sensing land-use data and socioeconomic statistical data using the economic–ecological coupling index method. Grossman [21] proposed that the two systems of economic development and ecological environmental pollution show an inverted U-shaped relationship. Liu [22] constructed the pressure–state–response (PSR) model framework of the coupling coordination mechanism between tourism economy and ecological environment, and evaluated the coupling coordination degree of the two. Zhao and Zuo [23,24] used the coupling coordination degree model to analyze the spatiotemporal evolution relationship between economic development and ecological environment, but it only considered the development trend of ecology and economy, while ignoring the feedback effect of ecological environment on economy. On the basis of the coupling coordination degree model, Zhang and Liu [25,26] introduced the geographically weighted regression model to analyze the spatiotemporal coupling coordination relationship between economic development and ecological environment, as well as its influencing factors. There were some limitations of the previous studies. Firstly, most of them studied the coupling coordination relationship between economic development and ecological environment from a static perspective, and few studied the relationship from a dynamic or spatial perspective. Secondly, the evaluation index system in most studies was similar, and the relationship between economic development and ecological environment could not be reasonably evaluated.

There are two ways to measure the quality of ecological environment. One is to select indicators from pollutant emission, greening rate, and waste treatment and recovery according to the pressure–state–response (PSR) model framework [27,28,29,30,31]. The second is to evaluate ecosystem service value through land-use change; in particular, the monetary evaluation method of ecosystem service value (ESV) has been widely used in the ecological field [32,33,34,35,36,37,38,39]. However, there are some limitations in the selection of quantitative indicators. First of all, sewage and waste emissions, energy consumption, carbon emissions, and other indicators are obtained through artificial statistical surveys, leading to statistical errors and other problems, with data often missing in small-scale research. Secondly, the change in land use/land cover is affected by physical and chemical properties, use mode, and other factors, and the impact on ecosystem service value is difficult to measure. In addition, the timeliness and accuracy of existing land-use datasets cannot meet the needs of timely and refined research.

With the development of remote sensing technology, multisource data fusion of remote sensing data, ecological data, and social and economic data should be strengthened [40,41,42]. Remote sensing observation technology has the advantages of wide observation range, short monitoring period, and large amount of information acquisition, and it is widely used in ecological environment monitoring [43,44]. Many scholars have constructed ecological indicators from the perspective of remote sensing [45,46,47]. The normalized difference vegetation Index (NDVI) [32,45,46,47,48,49] is one of the most commonly used ecological remote sensing indicators; it is used to calculate vegetation coverage, which can better reflect vegetation changes. The enhanced vegetation index (EVI) [32,49] is an improvement of NDVI, and it is superior to NDVI in reducing background and atmospheric effects and saturation problems. The normalized difference water index (NDWI) or modified normalized difference water index (MNDWI) [48,49,50,51] is sensitive to the recognition of surface water, and it is often used to extract surface water. Building and bare land can be identified and extracted using the normalized difference building index (NDBI) and bare soil index (BSI) or a composite index of building and normalized difference bare soil Index (NDBSI) [48,49,50,51]. Due to the influence of various complex factors, it is not enough to use a single ecological indicator to evaluate the status of the ecosystem; instead, comprehensive indicators are needed.

This paper selects Wuhan, an important central city in the Yangtze River economic belt, as a case study to analyze the interaction between ecology and regional economic development. In terms of geographical location, Wuhan is located at the intersection of the Yangtze River economic belt and the Beijing–Guangzhou development axis. It is a city in Central China with unique geographical advantages. With the acceleration of urbanization, the conflict between the natural ecosystem and social economic system is becoming more and more serious. Therefore, it is necessary to explore the interaction mechanism and evolution law between ecological environment and economic development. Our work mainly contributes in two ways:(1)A comprehensive evaluation index system of urban economy and ecological environment is constructed. The comprehensive ecological index system includes the ecological carbon sequestration index and land-cover index. The city economy comprehensive index system is constructed from three aspects: the economic development level, the economic structure, and the economic efficiency. The index weight is calculated by combining subjective and objective methods.(2)The interaction between ecological environment and urban economy in Wuhan city is analyzed from dynamic and spatial perspectives.

## 2. Study Area

Wuhan is a central city in central China, located at the intersection of the Yangtze River Economic Belt and the Beijing–Guangzhou development axis (see Figure 1). With an urban pattern of “two rivers and three towns” and convenient transportation of the thoroughfare of nine provinces, Wuhan has laid a regional leading foundation for the construction of ecological civilization. There are many rivers and lakes in Wuhan, including East Lake, the largest urban lake in China. Abundant ecological resources, such as wetlands, forests, and farmland, constitute a complete ecological system, forming the urban ecological characteristics of Wuhan, where land and water are interlaced, and human and water are interdependent.

## 3. Materials and Methods

### 3.1. Assessment Index System

This paper followed the principle of comprehensive and hierarchical index selection. It not only considered the availability of economic data of each administrative region in Wuhan, but also combined these data with remote sensing data to ensure their objectivity. The evaluation index system is shown in Table 1.

China’s economy has developed from a high-quality development stage to a high-speed growth stage [8]. Throughout the literature, urban economic indicators have been constructed by considering the quality of life, with more balanced, fuller, and greener development [9]; therefore, this article, from the aspects of economic development, economic structure and economic efficiency, selected 10 economic indicators, while Four ecological indices were selected from the aspects of ecological carbon sequestration and land cover.

(1)Economic development level index

GDP is an important indicator that reflects the regional economic situation and development level. Industrial GDP reflects the new added value of industrial enterprises in the production process. The disposable income of urban residents refers to the income that urban residents can freely control, while the per capita index can better reflect the living standard of people in the region.

(2)Economic structure index

The characteristics of China’s high-quality economic development include six aspects, one of which is the significant increase in tertiary industry’s contribution to economic growth. In recent years, the total proportion of the secondary industry and the tertiary industry in Wuhan has increased, especially the growth rate of the tertiary industry, accounting for more than 60%. Therefore, the proportion of the secondary industry, the proportion of the tertiary industry, the growth rate of the tertiary industry, and the growth rate of the gross national product are important indicators reflecting the economic structure of Wuhan city.

(3)Economic efficiency index

The selection of economic efficiency indicators mainly considers capital, resources, and other elements; in view of the availability of data at the district and city level, three indicators were selected: energy consumption per unit GDP, fixed asset investment of the whole society, and total retail sales of consumer goods.

(4)Ecological carbon sequestration index

The amount of organic matter accumulated in the unit of net photosynthetic productivity of plants is called net primary productivity (NPP). It represents the result of the interaction between plant biological characteristics and external environmental factors. As a key parameter of the terrestrial ecological process, NPP is an indispensable part of understanding the process of the terrestrial carbon cycle. It is also an important index to evaluate the structure and function of the ecosystem and the population carrying capacity of the biosphere.

The annual NPP is derived from the sum of all-day net photosynthesis (PSN) in a given year. The PSN value is the difference between total primary productivity (GPP) and total organic matter (MR) to maintain respiration. The calculation formula is as follows:(1)$$\mathrm{NPP}=\overset{n}{\underset{i=1}{\sum }}\mathrm{PSN}.$$
(2)PSN=GPP−MR. 

(5)Land-cover index

The vegetation, architecture, and water body were selected as the indicators of land cover, and the normalized difference vegetation index (NDVI) was used to reflect the ecological status. In this paper, NDBI and NDWI were innovatively introduced into the ecological index system. The normalized difference building index (NDBI) was used to reflect the urban building coverage, and the normalized difference water index (NDWI) was used to reflect the water coverage. The calculation formulas of NDVI, NDBI, and NDWI are as follows:(3)$$\mathrm{NDVI}=\frac{\left(\mathrm{NIR}-\mathrm{Red}\right)}{\left(\mathrm{NIR}+\mathrm{Red}\right)},$$
(4)$$\mathrm{NDBI}=\frac{\left(\mathrm{MIR}-\mathrm{NIR}\right)}{\left(\mathrm{MIR}+\mathrm{NIR}\right)},$$
(5)$$\mathrm{NDWI}=\frac{\left(\mathrm{Green}-\mathrm{NIR}\right)}{\left(\mathrm{Green}+\mathrm{NIR}\right)},$$
where NIR and MIR represent reflectance at the near-infrared and mid-infrared bands, while red and green represent reflectance at the red and green bands, respectively. The NDVI index, also known as the biomass index, is closely related to the transpiration and photosynthesis of plants; the combination of MIR and NIR constitutes the NDBI index. NDBI is mainly based on the high reflectivity of urban construction land in the TMS band. The research on urban land is often combined with NDVI; the combination of green and NIR constitutes the NDWI index. Due to the large number of rivers and lakes in Wuhan, the impact of water bodies on the ecological environment cannot be ignored. Therefore, the NDWI index was introduced to reflect the water coverage in Wuhan. The three indices were normalized, with values in the range of [−1, 1].

### 3.2. Data Processing and Sources

The data of economic indicators came from the statistical yearbook of Wuhan from 2014 to 2020 and the statistical data of various districts in Wuhan. The ecological index data were calculated from Landsat-8 and MODIS remote sensing satellite data, and the remote sensing images were obtained from the Google Earth Engine (GEE). The MODIS satellite provides annual net primary productivity (NPP) data with a spatial resolution of 500 m × 500 m. In addition, this paper also used the vector map of the Wuhan administrative division. The vector data came from the national geographic information resources directory service system. Some data of Hannan in 2016 and 2017 were missing; thus, the average annual growth rate method was used to make up the difference.

### 3.3. Index Weight Calculation

The method of combining subjective and objective indices was used to calculate the index weight, whereby the analytic hierarchy process (AHP) was used to calculate the subjective weight, and the entropy method was used to calculate the objective weight. Firstly, the subjective weight and objective weight of each level index were calculated according to the level of index division. Then, the subjective weight and objective weight were combined to obtain the comprehensive weight. Lastly, the scores of each index were calculated according to the comprehensive weight.

(1) Subjective weight calculation using AHP

(a)Constructing the first-order index judgment matrix

The discrimination matrix is established by pairwise comparison of the ratio of the importance of factor *i* to factor *j*.

(b)Consistency test


(6)
$$CI=\frac{\lambda_{max}-n}{n-1}.$$



(7)
$$CR=\frac{CI}{RI}.\;$$


(c)Calculating the weight proportion of each level of indicators

Construction of judgment matrix: (8)$$C_{j}=\frac{1}{n}\overset{n}{\underset{j=1}{\sum }}\frac{a_{ij}}{\sum_{k=1}^{n}a_{kj}};\left(i=1,2,\cdots ,n\right).$$

(2) Objective weight calculation by entropy method

(a)Calculation of standardized matrix

The range standardization method was used for processing, and the original data were transformed into a unified dimensional value by linear transformation. The calculation method was as follows:(9)$$b_{ij}=\frac{x_{ij}-min\left(x_{ij}\right)}{max\left(x_{ij}\right)-min\left(x_{ij}\right)}.$$

(b)Calculation of probability matrix *P*



(10)
$$p_{ij}=\frac{b_{ij}}{\sum_{i=1}^{m}b_{ij}}.$$



The sum of the probabilities of each index was guaranteed to be 1.

(c)Calculation of information entropy



(11)
$$h_{j}=-{\left(\ln n\right)}^{-1}\overset{m}{\underset{i=1}{\sum }}p_{ij}\ln p_{ij}.$$



When $p_{ij}=0$ and the $p_{ij}\ln p_{ij}=0$, then $0\leq h_{j}\leq 1$.

(d)Calculation of the information utility value

Greater information entropy indicates less information, which needs to be converted into information utility value. A greater information utility value corresponds to more information. The calculation formula of information utility value is as follows:(12)$$d_{j}=1-h_{j}.$$

(e)Calculation of the entropy weight of each index

The entropy weight of each index ($W_{j}$) is obtained as follows:(13)$$W_{j}=\frac{d_{j}}{\sum_{j=1}^{n}d_{j}}.$$

(3) Comprehensive weight of indicators

According to the first two steps, the weights under subjective and objective weighting methods were obtained, and the following formula was used to calculate the subjective and objective comprehensive weights:(14)$$K_{j}=\frac{\sqrt{C_{j}W_{j}}}{\sum_{j=1}^{n}\sqrt{C_{j}W_{j}}}.$$

(4) Comprehensive score of indicators
(15)$$S=\overset{n}{\underset{j=1}{\sum }}K_{j}\times score_{j},$$
where *n* is the number of composite indicators, from which the scores of economic comprehensive index and ecological comprehensive index can be obtained. The results are shown in Table 2.

Among the indicators of economic development level, the weight of per capita industrial GDP was the largest, reaching 54%, followed by 24% for per capita GDP and 22% for urban residents’ disposable income. This shows that, among the three indicators affecting the level of economic development, the per capita industrial GDP was more representative and contained more information.

Among the four indicators of economic structure, the proportion of the tertiary industry, the proportion of the secondary industry, and the growth rate of the tertiary industry were weighted as 35%, 33%, and 25%, respectively, considerably greater than the GDP growth rate, indicating their great impact on the economic structure.

Among economic efficiency indicators, the weight of energy consumption per unit GDP was the largest, reaching 58%, followed by 21% for social fixed assets investment and 21% for total retail sales of social consumption.

Among the comprehensive evaluation indicators of urban economy, the indicators of the economic development level, economic structure, and economic efficiency accounted for average weights of 30%, 38%, and 32% respectively.

In the comprehensive evaluation index of the ecological environment, the weight of the ecological carbon sequestration index was 70%, and the weight of the land-cover index was 30%. Among the land-cover indicators, the weights of NDVI, NDBI, and NDWI were 67%, 14%, and 19%, respectively. There were some differences in the weight of each index. Among them, the weight of the NDVI index was the highest with a larger amount of information, while the weights of NDBI and NDWI were smaller.

### 3.4. Elastic Coefficient Method

In order to explore the process of economic development in Wuhan City at the cost of consumption of ecological environment, this paper, on the basis of the Tapio decoupling model, selected 2014 as the base period to discuss the dynamic relationship between economic development and ecological environment from a relatively long-term perspective.

Generally speaking, the decoupling index takes GDP as an exogenous variable and energy consumption or carbon emissions as its external variable.

As explained variables, the degree and direction of decoupling are described. In this paper, the opposite of the eco-environmental comprehensive evaluation index was defined as the ecological loss, which was taken as the explanatory variable, and the comprehensive evaluation index of economic development was taken as the driving variable. Among them, since the comprehensive evaluation index of the ecological environment is a positive index, we used the opposite of the ecological environment index to modify the decoupling model.

According to domestic research, it is difficult to reflect the influence of relevant technology or policy on the “decoupling” trend over a long period from the data measurement of adjacent years; thus, the use of the link ratio form of the decoupling index is more in line with China’s national conditions. With the understanding of the concept of decoupling, the Tapio method was adopted to calculate the decoupling index of the relative base period year by year.
(16)$$ICDE_{t}=\frac{\frac{-\left(Y_{t}-Y_{0}\right)}{Y_{0}}}{\frac{\left(X_{t}-X_{0}\right)}{X_{0}}}=\frac{\frac{\Delta Y}{Y_{0}}}{\frac{\Delta X}{X_{0}}},$$
where ∆*Y* is the change in the comprehensive evaluation index of ecological environment, *X* is the change in the comprehensive evaluation index of urban economic development, *t* is the current period, and 0 is the base period. According to the definition of the Tapio decoupling model, combined with the calculation of the ecological environment index and urban economic comprehensive index in this paper, the decoupling index was divided as shown in Table 3.

### 3.5. Spatial Autocorrelation Model

A spatial weight matrix of 13 districts in Wuhan is constructed to represent the proximity relationship between each administrative district in Wuhan. 0 means that the two administrative districts are not adjacent in space, 1 means that the two administrative districts are adjacent in space.
(17)$$cw_{i,j}=\{\begin{array}{l} 0,\;\;i\;\mathrm{and}\;j\;\mathrm{are}\;\mathrm{not}\;\mathrm{adjacent}\\ 1,\;\;i\;\mathrm{and}\;j\;\mathrm{are}\;\mathrm{adjacent} \end{array}.$$

1. Moran’s I test 

(1) Global Moran’s I test

The global Moran’I is calculated using the following formula:(18)$$I=\frac{n}{S_{0}}\times \frac{\sum_{i=1}^{n}\sum_{j=1}^{n}\left(y_{i}-\bar{y}\right)\left(y_{j}-\bar{y}\right)}{\sum_{i=1}^{n}{\left(y_{i}-\bar{y}\right)}^{2}},$$
(19)$$S_{0}=\overset{n}{\underset{i=1}{\sum }}\overset{n}{\underset{j=1}{\sum }}w_{ij},$$
where *n* is the total number of administrative districts in Wuhan, which are the attribute values of the *i* and *j* administrative district units in Wuhan, *I* is the mean value of the unit attributes of all administrative districts in Wuhan, and *S* is the spatial weight value.

(2) Local Moran’s I test
(20)$$I_{i}=\frac{z_{i}}{s^{2}}\times \overset{n}{\underset{j\neq i}{\sum }}w_{ij}z_{j},$$
where $z_{i}=y_{i}-\bar{y}$, $z_{j}=y_{j}-\bar{y}$, $s^{2}=\frac{1}{n}\overset{n}{\underset{i=1}{\sum }}{\left(y_{i}-\bar{y}\right)}^{2}$, and $w_{ij}$ is the value of spatial weight. 

For the local Moran’s I test, four situations may occur: high/high region (HH), low/low region (LL), high/low region (HL), and low/high region (LH). The corresponding test value ranges are shown in Table 4.

## 4. Results

### 4.1. Analysis of Dynamic Relationship between Economic Development and Ecological Environment

The decoupling index of economic and ecological environment in Wuhan city from 2015 to 2020 was calculated, as shown in Table 5.

The results of decoupling index of Wuhan from 2014 to 2020 are described below. Throughout the whole research period, most of the decoupling indices of Wuhan’s economy and ecological environment were in a strong decoupling and weak decoupling state, i.e., the ecological loss had little relationship with economic development, indicating that Wuhan did not develop its economy at the expense of the environment, while the ecological environment and economic growth were in a relatively ideal state.

From the analysis of the decoupling relationship, there were development differences and spatial heterogeneity. Wuchang, Jianghan, Qiaokou, and Jiangxia were all in the most ideal state of “strong decoupling” from 2014 to 2020, indicating that Wuchang, Jianghan, Qiaokou, and Jiangxia were in the process of urban economic development, while the ecological environment was also optimized. The relationship between the two was realized, and the city developed in a green and low-carbon direction.

Jiang’an was only in a relatively ideal “weak decoupling” state in 2016, but in the most ideal “strong decoupling” state in other years, indicating that Jiang’an had a better green development and immediately mobilized the development direction after 2016. In line with urban economic development, the ecological environment was also optimized, and green and low-carbon development was continued.

From 2014 to 2020, Hongshan transformed its development from a “weak decoupling” to “strong decoupling” state. Hongshan reached a “strong decoupling” state in 2017 and then a “weak decoupling” state until 2020.

Qingshan was in a state of weak decoupling from 2014 to 2020. Qingshan is an important industrial town in Central China. Considering the development characteristics of the region, Qingshan should continue to promote green development, and further strengthen the pollution control and environmental protection in this heavy chemical industry city during the “14th Five Year Plan”.

The Wuhan Economic Development Zone (Hannan) basically maintained the best state of “strong decoupling”; it actively changed its policy and continued to maintain an optimal “weak decoupling” state since 2016.

Dongxihu was in a state of “strong negative decoupling” in 2016, which means that the ecological environment was declining, while the urban economic level is also regressing. According to the environmental bulletin of Dongxi Lake in 2016, it was found that the water quality of some lakes decreased slightly and did not reach the category of functional zones. At the same time, the government immediately made policy changes to strengthen the construction of the sewage collection pipe network and urban sewage treatment plant. After carrying out the purification project, the decoupling state of the area turned into an ideal state, which also shows that the policy response was positive, which can provide new ideas for the development of other regions.

Huangpi was also in the state of “strong negative decoupling” in 2016. In 2017, Huangpi actively promoted the feedback of central environmental protection supervision and the rectification of outstanding environmental problems assigned by provinces and cities to solve the outstanding problems reflected by the masses. After the orderly development of key environmental protection work, it turned into a “strong decoupling” state.

Xinzhou was in a state of “recession decoupling” from 2015 to 2016. In this regard, Xinzhou organized the construction of infrastructure for comprehensive rural environmental management in 2015 and actively changed the development direction. It subsequently improved in the direction of “strong decoupling”.

Hanyang was in a relatively ideal state before 2018. It was in recession decoupling state for 20 years despite a short recovery in the 19th year.

Caidian also responded to the policy call and gradually changed from strong negative decoupling to weak decoupling in the early years, which also reflected the promotion of a green development process in Wuhan.

According to the above results, from 2014 to 2020, from the perspective of a dynamic relationship, Wuchang, Jianghan, Qiaokou, and Jiangxia presented the best mutual development of urban economy and ecosystem environment among the 13 districts of Wuhan city.

### 4.2. Analysis of Spatial Interaction between Economic Development and Ecological Environment

Table 6 shows the single-variable global Moran’s I test results of the ecological comprehensive index. The results in each year were significantly positive, indicating that the ecological comprehensive index had a spatial effect.

As seen in Moran’s I scatter diagram (Figure 2), from the perspective of local correlation, the scatter points were concentrated in the first and third quadrants, denoting the high/high region (HH) and low/low region (LL), respectively, indicating that similar values were clustered and spatially correlated.

In order to show the test results more clearly, the ecological level LISA map of Wuhan city was drawn. According to Figure 3, there were only high/high areas, low/low areas, and insignificant results. Since 2014, only Hongshan District and Dongxihu District were high/high areas with high ecological level, whereas Jianghan District, Jiang’an District, Qiaokou District, Qingshan District, Hanyang District, and Wuchang District were low/low areas with a low ecological level. Jiangxia District and Hannan District remained insignificant, whereas Xinzhou District changed in some years. For example, Xinzhou District was not significant in 2014 and 2015, but became an HH area after 2015.

According to the quartile map of Wuhan city in Figure 4, Huangpi District, Xinzhou District, and Jiangxia District in 2014 and 2015 belonged to the first ecological level, Caidian district, Hannan District, and Hongshan District belonged to the second ecological level, and Dongxihu District, Jiang’an District, and Wuchang District belonged to the third ecological level. Hankou district, Qingshan District, and Hanyang District belonged to the fourth ecological level. In 2015 and 2016, Dongxihu District rose from the third ecological level to the second ecological level. In 2016, Hannan District was briefly lowered to the third ecological level before being restored to the second ecological level in the following year. In 2015, Hongshan District was briefly reduced to the third ecological level, before being restored to the second ecological level in the following year.

Firstly, in terms of the overall development of each district, the ecological level of Wuhan was generally “high in the periphery and low in the center”, which is contrary to the trend of the economic level, consistent with Krugman’s “center–periphery” theory, indicating that Wuhan is still trying to achieve the coordinated development of economy and ecology. Secondly, some of the outer districts had better ecological advantages, such as Huangpi District, Xinzhou District, Jiangxia District, Hannan District, and Caidian District. Hongshan District features the East Lake high-technology development zone, exhibiting ecological and geographical advantages with a high development potential.

## 5. Discussion and Policy Recommendations

This paper quantified the interaction between economic development and the ecological environment. According to previous studies, it is feasible to use remote sensing technology to monitor and evaluate the urban ecological environment [40,41,42,43]. In our study, we constructed three spectral indices, namely, NDVI, NDWI, and NDBI, as well as the NPP as a basic-level index, and we combined subjective and objective indices to calculate their weight, so as to integrate the quantitative factors. The relationship between these indicators and the ecological environment has been confirmed by studies. NDVI and NDWI are positively correlated with the ecological environment, while NDBI is negatively correlated with the ecological environment [52]. The administrative area of Wuhan was divided into high-ecological-value areas and low-ecological-value areas. Overall, the decoupling index of both areas showed an obvious difference. This difference was mainly reflected in the opposite trend over time, indicating “strong decoupling” since 2016. On 26 May 2016, Wuhan Municipal People’s Government issued a report on the establishment and improvement of the ecological compensation mechanism, confirming the rationality of our research results. The interaction between urbanization and the ecological environment is a dynamic process with a curvilinear change from imbalance to balance [53]. From the perspective of spatial heterogeneity, the ecological level of Wuhan generally presented an ecological layout of “high in the periphery and low in the center”, with the outer urban areas having high ecological value and the central city having low ecological value. The research of Jin et al. [54] showed that the development resistance of Wuhan presents a three-dimensional “inverted pyramid” spatial state. The ecological level of each region changed dynamically, but generally followed the layout according to “center and periphery”. This mechanism of action needs to be further studied.

According to the empirical analysis results of this paper, combined with the relationship between ecological environment and economic development, as well as the current rapid development of Wuhan, aiming at the urbanization process and carbon emission control and other factors, we provide the following suggestions:(1)Promote green development through local policy. We should continue to push the urbanization process forward while combining the development differences and spatial heterogeneity of different regions. Local governments should improve the efficiency of urban infrastructure, such as public systems, and reduce carbon emissions.(2)Adhere to green production and improve urban ecological environment. While urban economic development is being achieved, the ecological environment should also be optimized. A relationship between these two elements can move the city in the direction of green and low-carbon development.(3)Reasonably control energy consumption and raise awareness of low-carbon environmental protection. Energy consumption is a mainstay; hence, we should advocate the low-carbon concept in economic development and life, maintaining a simple and low-carbon consciousness. At the same time, with the continuous improvement of the technical level, we should pay attention to and provide targeted support for projects with less pollution and low energy consumption, such as transformation and marketing.(4)Promote the construction of wetland culture and grasp the ecological advantages. This can be achieved by protecting the integrity of wetlands, maintaining and increasing carbon sinks, building Wuhan’s “Wetland City” business card, and building an international wetland city.

Since China’s “double-carbon” development strategy was put forward, a series of related policies and action guidelines have been formulated, and various regions have successively made corresponding development goals and plans, as well as formulated specific implementation plans. However, the characteristics of population, finance, and natural resources differ according to region, along with the ability to implement policies. Therefore, from a policy point of view, how to implement emission reduction and increase foreign exchange in different regions is very important. Not only do we need to formulate corresponding policies according to local conditions, but we also need to explore various performance evaluations of the specific implementation of policies so as to better implement the goal of double-carbon development.

## 6. Conclusions

In this paper, the decoupling elastic coefficient method and spatial autocorrelation model were used to analyze the ecological environment and economic development in Wuhan, allowing several conclusions to be drawn. According to the analysis of the Tapio decoupling elasticity index, the ecological environment and economic growth of Wuhan were in a relatively ideal decoupling state throughout the research period, indicating that some regions of Wuhan paid equal attention to urban green sustainable development and economic and social development during this period. The ecological level index had a spatial effect, whereby improving the ecological level was also helpful in promoting economic development. Huangpi District, Xinzhou District, Jiangxia District, Hannan District, Caidian District, and Hongshan District have superior location and ecological advantages with high development potential. Considering the overall development of each district, Wuhan is still in the process of realizing the coordinated development of economy and ecology.

In this paper, when discussing the decoupling relationship between urban economy and ecological environment, the opposite of the comprehensive evaluation index of ecological environment was defined as ecological loss, which was used as the explained variable, while the comprehensive evaluation index of economic development was used as the driving variable. Therefore, the interpretation of the original decoupling model was the opposite. This approach is an improvement on the original Tapio decoupling model and makes up for the shortcomings of existing research methods. Secondly, compared with the static coupling relationship research, the Tapio decoupling coefficient could reflect the elasticity of ecological environmental indicators in the process of urban economic change, as well as reflect the dynamic trend of change. In addition, this study provides more possibilities for the application of economic data and remote sensing data, showing that it is necessary to apply remote sensing technology to economic and ecological fields.

This paper had some limitations. Firstly, our improved method of the Tapio model is reasonable at present, but whether it is applicable to other situations needs to be further discussed. Secondly, the study period was relatively short, and 2015 tended to reflect a state of decoupling. However, it is uncertain whether the state of the period before 2015 was cyclical or not, as well as the impact of policies. Lastly, the data had limitations. The data scale was 500 m, but the impact of different scales on the experimental results was not compared. Therefore, in subsequent research, a longer research period can be considered, and regional differences in the ecological level of Wuhan can be considered to conduct zoning modeling and multiscale modeling of Wuhan.

## Figures and Tables

**Figure 1 ijerph-19-10022-f001:**
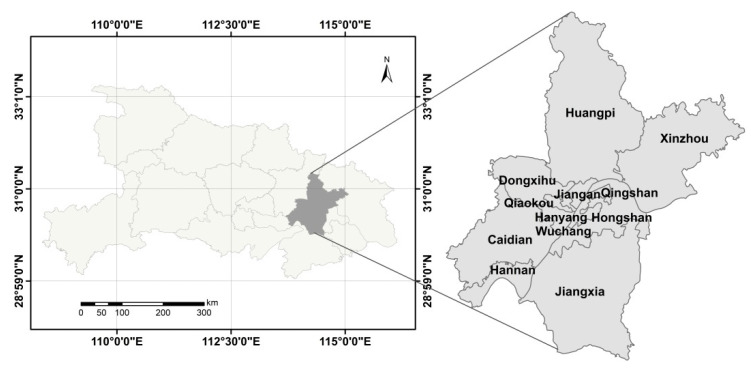
Location of study area.

**Figure 2 ijerph-19-10022-f002:**
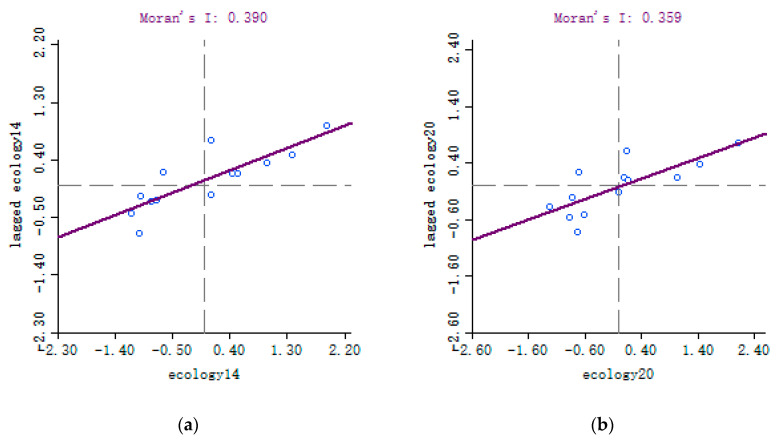
Moran’s I scatterplot of ecological level in Wuhan City. (**a**) 2014. (**b**) 2020.

**Figure 3 ijerph-19-10022-f003:**
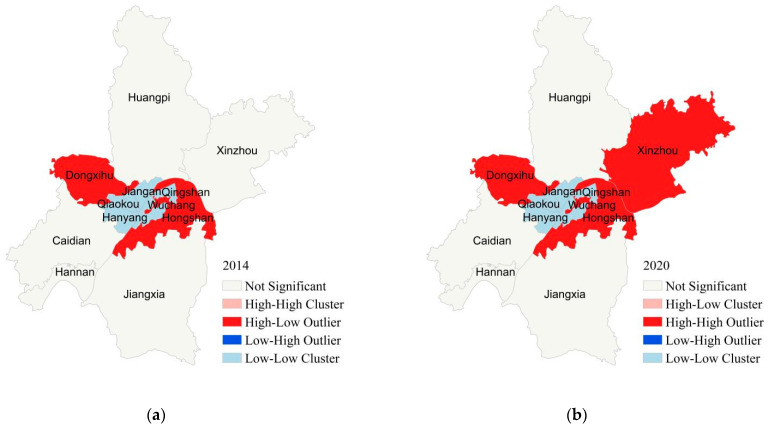
LISA cluster map of ecological level in Wuhan. (**a**) 2014. (**b**) 2020.

**Figure 4 ijerph-19-10022-f004:**
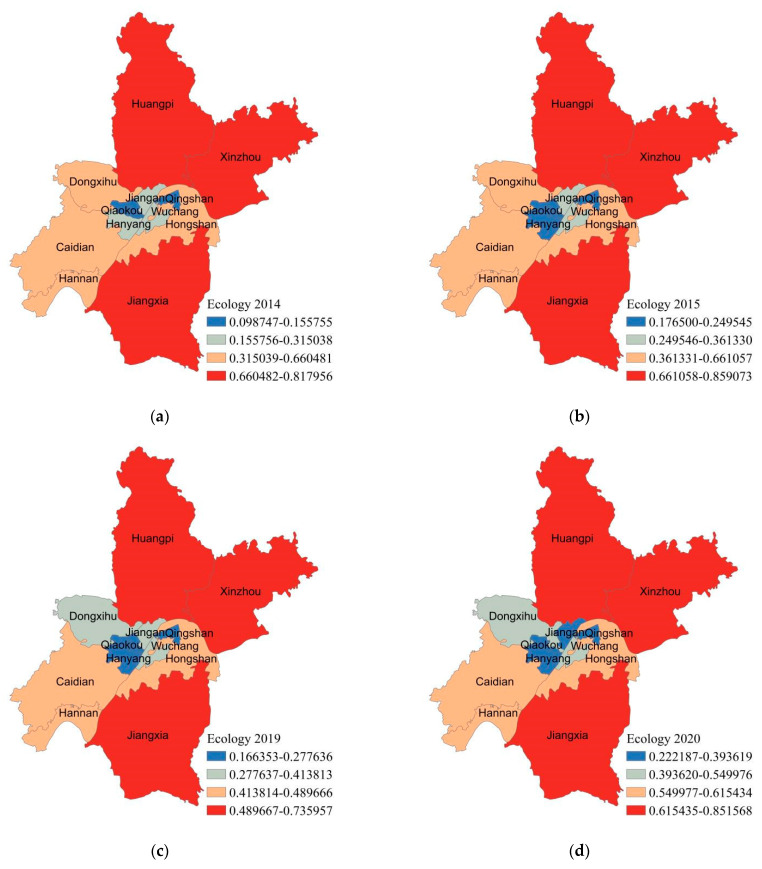
Quartile map of ecological level of Wuhan city. (**a**) 2014. (**b**) 2015. (**c**) 2019. (**d**) 2020.

**Table 1 ijerph-19-10022-t001:** Evaluation index system of economy and ecological environment in Wuhan.

Target Layer	Criterion Layer	Index Layer	Symbol
Comprehensive evaluation index *X* of urban economic development	Economic development level *X*_1_	Per capita GDP (10,000 CNY/person)	*X* _11_
		Per capita gross industrial product (10,000 CNY/person)	*X* _12_
		Annual disposable income of urban residents (CNY)	*X* _13_
	Economic structure index *X*_2_	Proportion of secondary industry	*X* _21_
		Proportion of tertiary industry	*X* _22_
		Growth rate of tertiary industry	*X* _23_
		GDP growth rate	*X* _24_
	Economic efficiency index *X*_3_	Energy consumption per unit of coal/GDP	*X* _31_
		Fixed assets investment of the whole society (100 million CNY)	*X* _32_
		Total retail sales of consumer goods (100 million CNY)	*X* _33_
Comprehensive evaluation index *Y* of ecological environment	Ecological carbon sequestration index *Y*_1_	Net primary productivity (NPP) (g/m^2^/year)	*Y* _11_
	Landover index *Y*_2_	Normalized difference vegetation index (NDVI)	*Y* _21_
		Normalized difference building index (NDBI)	*Y* _22_
		Normalized difference water index (NDWI)	*Y* _23_

**Table 2 ijerph-19-10022-t002:** Weights of indices.

Index Hierarchy	Index	Symbol	Subjective Weight	Objective Weight	Comprehensive Weight
**First-level index**	Indicators of economic development level	*X* _1_	0.21	0.38	0.30
	Economic structure index	*X* _2_	0.55	0.24	0.38
	Economic efficiency index	*X* _3_	0.24	0.38	0.32
	Ecological carbon sequestration index	*Y* _1_	0.67	0.74	0.70
	Land-cover index	*Y* _2_	0.33	0.26	0.30
**Basic-level index**	Per capita GDP	*X* _11_	0.25	0.23	0.24
	Per capita gross industrial product	*X* _12_	0.50	0.57	0.54
	Per capita disposable income of urban residents	*X* _13_	0.25	0.20	0.22
	Proportion of secondary industry	*X* _21_	0.23	0.39	0.33
	Proportion of tertiary industry	*X* _22_	0.23	0.45	0.35
	Growth rate of tertiary industry	*X* _23_	0.40	0.13	0.25
	GDP growth rate	*X* _24_	0.14	0.03	0.07
	GDP energy intensity	*X* _31_	0.54	0.59	0.58
	Investment in social fixed assets	*X* _32_	0.30	0.14	0.21
	Total retail sales of consumer goods	*X* _33_	0.16	0.27	0.21
	NPP	*Y* _11_	1.00	1.00	1.00
	NDVI	*Y* _21_	0.55	0.77	0.67
	NDBI	*Y* _22_	0.16	0.11	0.14
	NDWI	*Y* _23_	0.29	0.12	0.19

**Table 3 ijerph-19-10022-t003:** Index and types of decoupling elasticity between Tapio urban economy and ecological environment.

Decoupling Type	∆X	∆Y	Decoupling Index
Declining connection	<0	<0	(0.8,1.2)
Growth connectivity	>0	>0	(0.8,1.2)
recessive decoupling	<0	<0	(1.2,+∞)
Strong decoupling	<0	>0	(−∞,0)
Weak decoupling	>0	>0	(0,0.8)
Weak negative decoupling	<0	<0	(0,0.8)
Strong negative decoupling	>0	<0	(−∞,0)
Negative decoupling of growth	>0	>0	(1.2,+∞)

**Table 4 ijerph-19-10022-t004:** Results of local Moran’s I test.

$z_{i}$	$\overset{n}{\underset{j\neq i}{\sum }}w_{ij}z_{j}$	$I_{i}$	Partition	Description
>0	>0	>0	HH	The ecological level of area *i* is high, and the ecological level of the surrounding areas is high
<0	<0	>0	LL	The ecological level of area *i* is low, and the ecological level of the surrounding areas is low
<0	>0	<0	LH	The ecological level of area *i* is low, while that of the surrounding areas is high
>0	<0	<0	HL	The ecological level of area *i* is high, and the ecological level of the surrounding areas is low

**Table 5 ijerph-19-10022-t005:** Decoupling index of urban economy and ecological environment in Wuhan city from 2015 to 2020.

Year	2015	2016	2017	2018	2019	2020
Wuchang	−1.28	−0.43	−0.04	−0.27	−0.03	−0.45
Jianghan	−3.3	−1.71	−1.33	−1.39	−0.95	−2.14
Jiang’an	−2.62	0.02	−0.1	−0.34	−0.1	−0.42
Hongshan	0.16	0.32	−0.23	0.1	0.11	0.17
Qiaokou	−14.29	−2.68	−1.66	−2.33	−1.14	−2.4
Qingshan	0.46	0.17	0.14	0.65	0.23	0.63
Hannan	−0.04	0.02	−0.02	−0.01	−0.01	−0.08
Dongxihu	−0.63	−0.05	−1.43	−0.21	0.01	−0.14
Huangpi	−0.95	−0.23	−0.64	−0.29	−0.02	−0.04
Xinzhou	46.1	3.34	−3.93	−1.28	−0.19	−0.1
Jiangxia	−0.06	−0.21	−0.14	−0.17	−0.15	−0.09
Hanyang	−0.16	0.53	−0.2	4.36	−0.13	4.07
Caidian	−0.08	−1.42	0.16	−1.05	0.42	0.14

**Table 6 ijerph-19-10022-t006:** Single-variable global Moran’s I test results of ecological index.

Particular Year	Moran’s I Test		
	*I*	*z*-Value	*p*-Value
2020	0.353632	3.896291	0.000098
2019	0.342439	3.794738	0.000148
2018	0.385259	4.182494	0.000029
2017	0.384535	4.179121	0.000029
2016	0.393197	4.254598	0.000021
2015	0.436339	4.643034	0.000003
2014	0.443326	4.705794	0.000003

## Data Availability

All sources were provided in the paper.

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
