# Peer review of "Study on the Spatial Interaction between Urban Economic and Ecological Environment—A Case Study of Wuhan City"

_ijerph, 2022, doi:10.3390/ijerph191610022_

Round 1

Reviewer 1 Report

This paper is interesting and should be taken into consideration for publication in IJERPH. However, before it is publishable the paper hasto be improved in several areas as follows

1) The English readability of the paper should be improved, especially the abstract where many unnecessary hyphens appear. There are also many sentences which do not scan well in English, see for instance page 11, line 378 where the word 'from' is used three times in the same sentence.

2) The introduction should better position this study in the international literature of urban economics and ecological environments. Authors have to say what this study brings new or what it complements in the existing literature.

Also, the literature review need to say more on how urban economics has impacted the environment or the urban quarters where poor people live. It can be given examples of poor people in African cities. Even in Eastern Europe there is the case of urban Roma people with studies made for Europe (Powell and Lever, 2017), or case studies in Hungary (see Mereine Berki et al.'s studies in journal Cities in 2021 and in 2020 in journal Geographica Pannonica on how a disadvantaged neighbourhood of a city was desegregated and brought stigma for the displaced people of an urban environment lacking proper sanitation), or examples of urban economics of the poor people in Slovakia and other countries, or in cities from countries of Asia, Latin America etc. Also, authors mentioned foreign exchanges as important for urban economics, but it can be given several examples of foreign direct investments (FDIs) as case studies. For instance, in a chapter published in a book at Routledge in 2018 entitled 'Foreign direct investment and social risk in Romania - progress in less-favoured areas' it is shown that although there are many national project for attracting FDIs in poor towns and regions FDIs have been attracted mainly by big and attractive cities. Similar examples like the above ones have to be given more in the introduction and discussions of this paper so that to better position this study at the international level and link it to other international studies.

3) The methods and the results sections are the strongest parts of the paper, but figure 3 and 4 should be made larger, otherwise readers cannot see the details presented in those figures.

4) There are no discussions in this paper. Authors have to link their specific results to what other similar or different results have been published by now. So, 3-4 new paragraphs are needed to be drawn as Discussions in the paper. 

5) The policy recommendations should be better placed at the end of the discussions, while conclusions should be expanded by presenting how the aims of the paper have been solved, the implications of this study at international level, and how other authors can complement the outcomes of thsi study.

6) The reference list is made of many Chinese references, which is good. I think more references are needed with examples from the urban economics and ecological environments of other urban studies in different states of the world. This is related to the above request of improving the literature review, see my point number 2.

Reviewer 2 Report

In this paper, authors use the data of Wuhan city to construct two indexes of urban economic development and ecological environment, and then try to analyze the spatial interaction between these indexes. Although the spatial analysis and regression method seems to be standard, there are still several important issues that haven’t been explained well in the current manuscript. The main comments are as follows:

Major comments:

1.       The construction of index system of urban economic development and ecological environment, the two key variables in this paper, looks arbitrary. Authors provided the conclusions of two papers to support their ideas. However, they didn’t provide enough explanation and discussion in the construction process of each index. This is important because the whole paper is based on these two indexes, authors need to give the sufficient evidence from the existing literature, or economic theories to explain why the chosen data in calculating the indexes and their method of index weight calculation are suitable in this study.

2.       The figures in pages 12 and 13 are too much and too small because authors add all 7 years’ (2014-2020) figures of Moran’s I scatterplot, Lisa cluster map and quartile map. I can’t find the significant difference among 7 figures. Authors may just keep several representative figures and explain the outcomes of other figures in words, or put some figures to the Appendix.

3.       The English language in this paper is not very fluent; authors should find some native English speaker to give a careful proofreading.  

Minor comments:

1.       There are some errors and typos in the paper. For example, the title of paper is “Study on the spatial interaction between urban economic and ecological environment”, it should be “urban economy”, instead of “urban economic”, according to the content of paper. Authors should check the whole paper to avoid such error or typo.

Round 2

Reviewer 1 Report

Authors have improved their paper but a minor revision is further needed on two aspects:

1) the final order of the sections should be: Discussions and policy recommendations, and then Conclusions. So just move conclusions to be the last section and merge discussions with policy aspects because discussions are quite short to stand a separate section. Also, in the discussions it would be good to connect some of the findings of international studies (Octavio et al, Basit el al) and Chinese studies (see references of Deng, or Chen, or even Liu etc) with the findings of this paper.

Still at the moment conclusions are short. They need to show the international implications of this study or in other words to say how the novelty of this study brought new aspects to what has been published by now. Also, conclusions should present how other authors can develop further the outcomes of this paper. Finally, the limitations of this study can be presented too in the conclusions.

2) A smaller remark is to re-write the lines 43-45 from bottom of page 1 and first line on page 2, because at the moment those two sentences look confusing on regional impact and not on urban impact. So instead of 'The studies of Powell....' just put 'Moreover, the studies of Powell and Lever[9] on Roma outsiders, and Cretan et al. [10] and Méreiné-Berki et al [11] on urban Roma confirms that marginalized residents could face strong negative community impact at urban level.'

Reviewer 2 Report

The authors have revised the paper well, and I am satisified with the current manuscript.
